# Cryopreserved Human Natural Killer Cells Exhibit Potent Antitumor Efficacy against Orthotopic Pancreatic Cancer through Efficient Tumor-Homing and Cytolytic Ability (Running Title: Cryopreserved NK Cells Exhibit Antitumor Effect)

**DOI:** 10.3390/cancers11070966

**Published:** 2019-07-09

**Authors:** Eonju Oh, Bokyung Min, Yan Li, ChunYing Lian, JinWoo Hong, Gyeong-min Park, Bitna Yang, Sung Yoo Cho, Yu Kyeong Hwang, Chae-Ok Yun

**Affiliations:** 1Department of Bioengineering, College of Engineering, Hanyang University, 222 Wangsimni-ro, Seongdong-gu, Seoul 04763, Korea; 2GeneMedicine Co., Ltd., Seoul 04763, Korea; 3GC LabCell 107, Ihyeon-ro 30beon-gil, Giheung-gu, Yongin-si, Gyeonggi-do 16924, Korea; 4Institute of Nano Science and Technology (INST), Hanyang University, 222 Wangsimni-ro, Seongdong-gu, Seoul 04763, Korea

**Keywords:** adoptive cell therapy, allogeneic natural killer cells, cancer immunology, pancreatic cancer, apoptosis

## Abstract

Pancreatic cancer is known to be highly aggressive, and desmoplasia-induced accumulation of extracellular matrix (ECM), which is a hallmark of many pancreatic cancers, severely restricts the therapeutic efficacy of both immunotherapeutics and conventional chemotherapeutics due to the ECM functioning as a major physical barrier against permeation and penetration. In the case of cell-based immunotherapeutics, there are several other bottlenecks preventing translation into clinical use due to their biological nature; for example, poor availability of cell therapeutic in a readily usable form due to difficulties in production, handling, shipping, and storage. To address these challenges, we have isolated allogeneic natural killer (NK) cells from healthy donors and expanded them *in vitro* to generate cryopreserved stocks. These cryopreserved NK cells were thawed to evaluate their therapeutic efficacy against desmoplastic pancreatic tumors, ultimately aiming to develop a readily accessible and mass-producible off-the-shelf cell-based immunotherapeutic. The cultured NK cells post-thawing retained highly pure populations of activated NK cells that expressed various activating receptors and a chemokine receptor. Furthermore, systemic administration of NK cells induced greater *in vivo* tumor growth suppression when compared with gemcitabine, which is the standard chemotherapeutic used for pancreatic cancer treatment. The potent antitumor effect of NK cells was mediated by efficient tumor-homing ability and infiltration into desmoplastic tumor tissues. Moreover, the infiltration of NK cells led to strong induction of apoptosis, elevated expression of the antitumor cytokine interferon (IFN)-γ, and inhibited expression of the immunosuppressive transforming growth factor (TGF)-β in tumor tissues. Expanded and cryopreserved NK cells are strong candidates for future cell-mediated systemic immunotherapy against pancreatic cancer.

## 1. Introduction

Pancreatic cancer is the fifth leading cause of death from cancer in the world, and the 5-year survival rate is a dismal 6% in Europe and the United States [1,2,3]. This is partly because conventional therapies, such as chemotherapy, show insufficient therapeutic efficacy against advanced stages [4]. Currently, gemcitabine monotherapy or gemcitabine plus nab-paclitaxel are the standard first-line treatment for metastatic pancreatic cancer [5,6]. However, overall survival benefits provided by either gemcitabine-based therapy are limited by the highly desmoplastic nature of pancreatic cancer, which prohibits drug penetration and dispersion into tumor tissues. 

Because of these limitations of conventional treatment methods, various immunotherapeutics, such as immune checkpoint inhibitors, cancer vaccines, or chimeric antigen receptor T cells [7], are currently being evaluated in clinical trials for the treatment of pancreatic cancer [8]. This direction has been taken in part because of the observation that higher levels of immune cells, such as natural killer (NK) cells, in circulation are associated with improved survival of pancreatic cancer patients, suggesting that boosting of host antitumor immunity by therapeutics may lead to better disease management [9,10,11,12]. NK cells are specialized components of the innate immune system that contribute to the first line of defense against viral infections and cancer [13,14,15]. NK cell-based cancer immunotherapy can be achieved by several different methods, such as activation of endogenous NK cells and adoptive transfer of *ex vivo*-expanded autologous or allogeneic NK cells [16,17,18,19,20]. One of the main advantages of NK cells over other immune cell-based therapeutics is that they can induce direct killing of target cells without any prior stimulation [21,22,23,24]. The cytolytic activity of NK cells is major histocompatibility complex (MHC)-unrestricted, and therefore plays an antitumor surveillance role distinct from that of T cells [25,26,27]. In addition, NK cells contribute to the induction of adaptive immune responses by secreting cytokines and chemokines [28]. Because of their strong promise, NK cells are currently being evaluated in various clinical trials for the treatment of human cancers [17].

Recently, development of allogeneic NK cells has gained much attention due to autologous NK cells from cancer patients often being functionally impaired during advanced stages of the disease [29,30,31]. The inhibition of autologous NK cells’ function is mediated by binding of self-human leukocyte antigen (HLA) class I molecules to killer cell immunoglobulin-like receptors (KIRs); a suppressive interaction that overrides activation signals [32,33,34]. In contrast, allogeneic NK cells have been shown to elicit potent antitumor efficacy due to induction of graft-versus-tumor effects [23,35,36]. Several recent clinical studies have shown that alloreactive NK cells with KIR-HLA mismatch can effectively kill tumor cells and inhibit recurrence of acute myeloid leukemia [37,38,39,40]. Allogeneic NK cells have also shown promising preliminary results against solid tumors, such as renal cell carcinoma, malignant melanoma, lung cancer, and hepatic cancer in recent clinical trials [18,19,20]. Despite these promising results, there are several major challenges that must be addressed for allogeneic NK cell therapy to be effective and practical. First, it must be possible to easily expand and maintain, *ex vivo*, a sufficient quantity of highly purified NK cells without contamination by other lymphocytes. Second, adoptively transferred NK cells must demonstrate migration and infiltration into tumor tissues while preserving their activity *in vivo*. The optimization of cryopreservation and thawing procedures for therapeutic NK cells can address some of these challenges. The cryopreservation of NK cells can enable long-term storage of NK cells at medical facilities, allowing sufficient quantities of cells to be readily available for patients.

In the present study, allogeneic human NK cells derived from KIR haplotype B healthy donor peripheral blood mononuclear cells (PBMCs) (MG4101) were generated by an improved method for large-scale expansion and cryopreservation of NK cells. These cryopreserved allogenic NK cells after thawing induced potent cytotoxic effects against human pancreatic cancer cells *in vitro* by expression of activating receptors, secretion of cytokines, and strong induction of apoptosis. Further, allogeneic human NK cells successfully inhibited tumor progression in a human pancreatic orthotopic tumor model, demonstrating that these NK cells are a promising candidate for future cell-mediated immunotherapy clinical trials for the treatment of pancreatic cancer.

## 2. Results 

### 2.1. Characterization of Ex Vivo-Large-Scale Expanded and Frozen NK cells

For translating NK cell immunotherapy to the clinic, it is essential that NK cells can be cryopreserved and thawed without functional impairment and exhibit similar activity as freshly isolated NK cells [41]. Currently, the viability and activity of NK cells are severely reduced immediately after thawing [42,43]. To address this need, we have expanded and cryopreserved NK cells to assess various effects of cryopreservation on NK cell activities. Allogeneic NK cells derived from CD3-depleted PBMCs of seven healthy donors were expanded by stimulating irradiated PBMCs in the presence of purified anti-human CD3 antibody (clone: OKT3) and interleukin (IL)-2. During a 3-week culture period, fresh NK cells were efficiently expanded, showing 4290.2 ± 3812.8-fold increases in NK cell count during this period (Figure 1A). 

Next, we tested the viability and activity of cryopreserved NK cells that had been thawed and cultured for three weeks. As shown in Figure 1B, NK cells preserved viability of 89.9 ± 4.5% after cryopreservation. In addition, the cell population from cryopreserved samples was composed of highly enriched CD3^−^CD56^+^ NK cells (98.7 ± 0.7%) with minimal contamination of CD3^+^ T cells (1.0 ± 0.7%), CD14^+^ monocytes (0.7 ± 0.7%), and CD19^+^ B cells (0.2 ± 0.2%) (Figure 1C). As NK cell function is finely regulated by the balance between activating and inhibitory receptors expressed on their surface, we assessed the expression levels of activating and inhibiting receptors on the surface of cryopreserved NK cells post-thawing (Figure 1D). Among activating receptors, CD16, NKG2D, NKp30, NKp46, and DNAM-1 were expressed at a higher level than inhibitory receptor, NKG2A, during expansion. The expression levels of two other activating receptors, NKG2C and NKp44, were largely unchanged compared to those of other activating receptors (CD16, NKG2D, NKp30, NKp46, and DNAM-1). In addition, the chemokine receptor CXCR3 was expressed. Based on these results and those previously published [44], cryopreserved NK cells post-thawing retain the phenotype of highly pure NK cell population. 

### 2.2. Characteristic Changes in NK Cells Co-Cultured with Cancer Cells

Antitumor efficacy of adoptively transferred NK cells requires the expression of various molecules to recognize and kill target cells [45]. To test their activity, NK cells were co-cultured with either an NK cell-susceptible cell line (K562) or pancreatic cancer cells (AsPC-1, MIA PaCa-2, Capan-1, and PANC-1) for 4 h and then assessed for direct cell killing activity and the expression level of cytokines as well as degranulation marker (CD107a). NK cell-susceptible K562 cells were used as a positive control for the analysis of pancreatic cancer cell susceptibility to allogeneic NK cells. As shown in Figure 2, K562 cells were found to be the most susceptible to NK cell lysis (84.4 ± 5.4%). MIA PaCa-2, Capan-1, and PANC-1 were highly sensitive to NK cell-mediated cytolytic activity (44.5 ± 9.3%, 48.7 ± 2.3%, and 47.4 ± 13.8%, respectively) in comparison to AsPC-1 (13.4 ± 7.8%), which was the most resistant toward NK cell-mediated lysis. In line with these results, the potent cytotoxic activity of NK cells was confirmed by analyzing the expression level of degranulation marker cluster of differentiation (CD)107a (14.7 ± 3.8%, 9.8 ± 3.2%, 18.5 ± 5.0%, 13.2 ± 4.5%, and 43.5 ± 6.6%), secretion of interferon (IFN)-γ (10.7 ± 3.1%, 10.0 ± 4.4%, 13.7 ± 2.5%, 9.4 ± 3.5%, and 40.5 ± 6.5%), and secretion of tumor necrosis factor (TNF)-α (10.1 ± 4.3%, 8.5 ± 5.5%, 12.5 ± 3.4%, 8.0 ± 3.6%, and 33.0 ± 11.1%) following co-culturing of NK cells with AsPC-1, MIA PaCa-2, Capan-1, PANC-1, and K562, respectively. 

NK cells recognize target cells by interacting with various inhibitory and activating receptors, which regulate NK cell-mediated lysis [46,47,48]. Thus, we compared the surface expression levels of various activating receptors on NK cells cultured with or without pancreatic cancer cells (MIA PaCa-2, AsPC-1, and Capan-1) at 24 h intervals (Figure 3A–C). Surprisingly, we detected a significant reduction in activating receptors (CD16, NKG2D, NKp30, NKp44, NKp46, DNAM-1, CD96, or CD161) after co-cultivation with cancer cells compared to control NK cells. The expression level of chemokine receptor (CXCR3) was also significantly altered by co-cultivation with cancer cells (Figure 3B). These results suggest that there are significant alterations to the surface expression levels of various activating receptors of NK cells following cell-to-cell contact with pancreatic cancer cells. 

To assess the roles of activating NK receptors, cytotoxicity assays with NK cells were performed in the presence of blocking antibodies (Abs) specific to NKp30, NKp44, NKG2D, and DNAM-1. As shown in Figure 4, blocking a single receptor induced minor inhibition of NK cell-mediated cytotoxicity. Importantly, blocking multiple receptors led to markedly higher levels of inhibition than individual single receptor blockage. Together, these results suggest that the cytolytic activity of expanded NK cells requires various activating receptors on NK cells for direct contact between NK cells and pancreatic cancer cells.

### 2.3. Cytotoxic Effect of NK Cells against Human Pancreatic Cancer Cell Lines

To evaluate the cancer cell killing effects of the ex vivo expanded and cryopreserved NK cells, 3-(4,5-dimethylthiazol-2-yl)-2,5-diphenyltetrazolium bromide (MTT) assays were performed using human pancreatic cancer cells (MIA PaCa-2 and PANC-1) following treatment with various E:T ratios of NK cells. As shown in Figure 5A, NK cells exhibited dose-dependent cancer cell-killing activity (*p* < 0.05, *p* < 0.01, or *p* < 0.001 versus phosphate-buffered saline (PBS) for MIA PaCa-2; *p* < 0.001 versus PBS for PANC-1). Furthermore, the NK cell-mediated killing of both MIA PaCa-2 and PANC-1 cells gradually increased from 48 h to 96 h post treatment. These results indicate that NK cells can elicit potent cytocidal effect against pancreatic cancer cells. 

### 2.4. Potent Induction of Apoptotic Cancer Cell Death by NK Cells

To test whether the cancer cell killing effect of the NK cells is mediated by the induction of apoptosis, annexin V/propidium iodide (PI) staining assay was performed. As shown in Figure 5B, NK cell-mediated induction of apoptosis was significantly enhanced in a dose-dependent manner (*p* < 0.001 versus PBS for MIA PaCa-2 cells, *p* < 0.01 or *p* < 0.001 versus PBS for PANC-1 cells, respectively). These results suggest that NK cells can elicit the anticancer effect via induction of apoptosis. 

### 2.5. Therapeutic Efficacy of NK Cells in an Orthotopic Pancreatic Tumor Model

Orthotopic tumor models are emerging as reliable *in vivo* preclinical models because of their close emulation of disease progression and tumor microenvironment [49,50]. To evaluate the therapeutic efficacy of the *ex vivo*-expanded NK cells *in vivo*, MIA PaCa-2 orthotopic pancreatic tumor-bearing mice were treated with NK cells (2 × 10^7^ cells via intravenous injection), IL-2 (250 U via intravenous injection), or gemcitabine (100 mg/kg via intraperitoneal injection). Gemcitabine was utilized as a positive control because of its common usage as the standard treatment for advanced and metastatic pancreatic cancer [6,51]. As shown in Figure 6A, NK cells elicited higher level of antitumor activity in respect to conventional chemotherapy option of gemcitabine. At 14 days after injection with NK cells, the luciferase signal from the orthotopic pancreatic tumors was significantly reduced in comparison to other groups, showing 7.2- or 6.9-fold higher tumor growth inhibition than PBS or IL-2, respectively (Figure 6B; *p* < 0.01). More importantly, NK cells displayed markedly greater antitumor efficacy than gemcitabine, showing 4.0-fold reduction in tumor burden in comparison to mice treated with gemcitabine (*p* < 0.01). 

### 2.6. Increased IFN-γ Expression and Decreased TGF-β Expression in Tumor Tissues Following NK Cell Treatment 

Previous studies reported that IFN-γ, a T helper (Th)1 cytokine, is a key contributor to the induction of antitumor immune response [52,53,54]. Therefore, we assessed the expression levels of IFN-γ in tumor tissues after treatment with NK cells. As shown in Figure 7, tumors treated with NK cells exhibited significantly higher levels of IFN-γ than those treated with PBS or gemcitabine (*p* < 0.001). Interestingly, NK cell-treated mice showed basal levels of IFN-γ in serum, similar to those observed in mice treated with PBS or gemcitabine, showing that NK cell-mediated induction of antitumor immune response was highly localized to tumor tissues. This localization may be beneficial, as high expression levels of serum cytokines can induce severe systemic side effects [55,56,57,58].

Several reports have shown that overexpression of immunosuppressive factors, such as transforming growth factor (TGF)-β in the tumor microenvironment can significantly diminish the potency of immunotherapeutics [59,60,61]. Thus, we next assessed the effect of each treatment on the inhibition of TGF-β expression in pancreatic tumor tissues and serum of mice following treatment with PBS, gemcitabine, or NK cells. As shown in Figure 7, both gemcitabine and NK cell treatment led to marked attenuation of TGF-β expression levels in tumor tissues compared to those observed in mice treated with PBS (*p* < 0.001). Together, these results demonstrate that NK cells can induce potent antitumor effects by simultaneously elevating expression levels of the antitumor cytokine IFN-γ and inhibiting expression of the immunosuppressive factor TGF-β in the tumor microenvironment. 

### 2.7. Histological and Immunohistological Analyses of Orthotopic Pancreatic Tumors Treated with NK Cells

To further investigate the therapeutic effects of NK cells, tumor tissues harvested at 3 days post-final treatment (PBS, gemcitabine, or NK cells) were histologically and immunohistochemically analyzed. As shown in Figure 8A, hematoxylin and eosin (H & E) and proliferating cell nuclear antigen (PCNA) staining revealed large areas of proliferating tumor cells in PBS-treated tissues, whereas moderate or extensive necrosis with markedly lower levels of proliferating tumor cells were observed in gemcitabine or NK cell-treated groups, respectively. In particular, NK cell-treated tumors showed larger necrotic areas and lower quantities of PCNA-positive proliferating tumor cells than tumors treated with gemcitabine. Importantly, human NK cell marker CD56 was detected in tumor tissues of mice treated with NK cells via systemic administration (Figure 8B), suggesting that allogeneic NK cells can localize to and infiltrate the tumor tissues. Terminal deoxynucleotidyl transferase dUTP nick end labeling (TUNEL) staining results, in support of PCNA staining, revealed that both gemcitabine and NK cells treatment led to induction of apoptosis in tumor tissues; with the NK cell-treated group showing the most robust induction of apoptosis. Together, these results show that systemically administered NK cells can successfully infiltrate desmoplastic pancreatic tumors to strongly induce apoptosis.

Next, we assessed by immunohistochemistry whether the intratumoral infiltration of allogeneic NK cells can promote infiltration of host NK cells to tumor tissues. As shown in Figure 8C, tumors of mice systemically treated with either gemcitabine or NK cells showed markedly higher levels of endogenous murine NK cell infiltration, suggesting that suppression of immunosuppressive TGF-β by either treatment may facilitate intratumoral infiltration of immune cells. Of note, the tumor tissues of NK cell-treated mice showed greater infiltration of endogenous murine NK cells than the adjacent normal tissues or tumor tissues of mice treated with gemcitabine. Together, these results suggest that either NK cell treatment-mediated elevation in antitumor cytokine (IFN-γ) or strong induction of apoptosis in tumor tissues may further improve localization and infiltration of murine NK cells.

### 2.8. Biodistribution Profile of Systemically Administered NK Cells 

The biodistribution profile and tumor-homing ability of transfused effector cells are major areas of interest for cell therapy. Typically, NK cells are detected in small numbers within advanced human neoplasms, suggesting that normally they do not efficiently infiltrate malignant tissues [62,63,64]. To evaluate whether systemically administered NK cells can effectively target and infiltrate tumor tissues, the biodistribution profiles of systemically administered NK cells were assessed by immunohistochemistry. Mice bearing MIA PaCa-2 orthotopic tumors were treated intravenously three times a week for 2 weeks with NK cells and then sacrificed 1, 2, or 7 days after the last injection. As shown in Figure 8D, there was a markedly higher number of systemically administered human NK cells in tumor regions than in the adjacent normal pancreatic tissues at day 1 post-injection. By day 7, NK cells were no longer detectable in tumor regions. In addition, NK cells were observed in the spleen, liver, lung, and kidney. In particular, NK cells were found in higher numbers in the spleen than in other organs at all time points. Collectively, these results demonstrate that NK cells efficiently home to tumor tissues and infiltrate them, thus resulting in tumor growth inhibition.

## 3. Discussion

Traditional chemotherapeutics have failed to substantially improve the survival rate of pancreatic cancer patients [8]. Given the poor clinical result of patients with pancreatic cancer, novel methods are required to introduce new treatment opportunities. Recently, research efforts have focused on the development of immunotherapeutics, such as various types of immune cells, against cancer [65,66,67,68,69]. Among these immune cells, allogeneic NK cells are particularly promising because donor-recipient incompatibility between KIRs on donor NK cells and KIR ligands on recipient tissues results in induction of potent anticancer immune responses [17,70]. In this study, we demonstrate the feasibility of adoptively transferring ex vivo expanded human NK cells derived from KIR haplotype B healthy donors.

Cryopreservation of NK cells would allow for the development of a readily accessible off-the-shelf product and production in batches, thus making the therapy more applicable to clinical use [41]. However, NK cells are known to be more sensitive to freezing and thawing than other lymphocytes [71], resulting in inferior rates of cell recovery and potency of cryopreserved NK cells upon thawing [72]. In the present report, we used freezing media for cell cryopreservation. We observed high percentage viability of the cryopreserved NK cells post-thawing (Figure 1B). Further, the cultured NK cells recovering from thawing were extremely pure populations of activated NK cells that expressed various activating receptors and a chemokine receptor (Figure 1C,D). These findings suggest that cryopreservation of NK cells under this particular composition of freezing media (in process of patent filing) did not affect viability or phenotype of the cells. 

The successful application of NK cell-based therapies in solid tumors faces several challenges [73,74]. One is that targeted tumors must be susceptible to NK cell-mediated cytotoxicity. NK cells recognize their target cells by a combination of signals from inhibitory and activating receptors [75]. Thus, susceptibility of target cells depends on the surface expression level of chemokines, inhibitory, and activating receptors or ligands by NK cells. The cryopreserved NK cells post-thawing showed robust cytokine production, induction of apoptosis, and potent cytolytic activity against pancreatic cancer cell lines (Figure 2 and Figure 5). Interestingly, these NK cells showed lower expression level of activating receptors after co-cultivation with pancreatic cancer cells (Figure 3). This finding is reminiscent of previous data regarding the downregulation of NKG2D on T cells upon their binding with tumor-associated ligands via cell contact-dependent trogocytosis, which may promote immune synapse formation through receptor-ligand interaction [76,77,78]. In addition, several studies have reported that many receptors undergo ligand-induced endocytosis followed by recycling or degradation [79,80,81]. Importantly, the cytotoxicity of NK cells was significantly inhibited when the cells were preincubated with multiple antibodies blocking the activating NK receptors (Figure 4). Based on our results, the interaction between NK activating receptors and cancer cells is integral to the cytolytic activity of NK cells. Moreover, NK cell treatment led to robust induction of apoptosis (Figure 5B and Figure 8A), which was supported by previous study demonstrating that NK cell-mediated expression of TNF superfamily proteins induces apoptosis by their engaging cell death receptors on target cells [17]. Our findings show that the cryopreserved NK cells post-thawing can elicit potent cytocidal effects against pancreatic cancer cells through induction of apoptosis, suggesting that cryopreservation did not affect NK cell function.

The tumor microenvironment possesses various types of immunosuppressive cells including myeloid-derived suppressor cells (MDSC), T regulatory (Treg) cells, and tumor-associated macrophages (TAM) [82]. Specifically, Treg- and MDSC-mediated expression of TGF-β a negative immune regulator, can impair NK cells’ ability to kill tumor cells [83]. In the current study, we found that systemic administration of allogeneic NK cells significantly attenuated TGF-β expression in tumor mass, suggesting that allogeneic NK cells may attenuate immunosuppressive burden in the tumor milieu (Figure 7). This phenomenon was likely mediated by the expression of human IFN-γ, which possesses cross-reactivity to its murine counterpart [83]. The interaction between human IFN-γ and murine IFN-γ receptors may induce the activation of endogenous immune cells and secretion of immunostimulatory factors, ultimately resulting in inhibition of TGF-β expression in the tumor microenvironment [84,85] Surprisingly, we observed TGF-β expression was also decreased in the tumor following administration of gemcitabine. This might be due to gemcitabine reducing the MDSC population that is capable of secreting TGF-β in the tumor milieu [86]. 

Other critical factors determining the overall efficacy of NK cell-based immunotherapies against solid tumor are their abilities to home and infiltrate into tumor tissues for induction of effector function [45]. To this end, our allogeneic NK cells showed good tumor-homing ability, as greater quantities of CD56-positive NK cells were detected in tumor tissues than other normal tissues (Figure 8B). Furthermore, these therapeutic NK cells also showed good tumor penetrating ability against desmoplastic pancreatic tumors, which are known to possess dense layers of tumor extracellular matrix: this observation is based on NK cells being evenly distributed through both peripheral and central tumor regions. This penetration ability of NK cells may be attributed to their being CXCR3-positive subsets (Figure 1D), as several reports have shown that CXCR3-positive NK cells efficiently infiltrate tumors [87,88,89]. Additionally, efficient infiltration of allogeneic NK cells was directly correlated with infiltration of endogenous NK cells (DX5-positive) and intratumoral IFN-γ expression (Figure 7 and Figure 8C), showing that allogeneic human NK cells can facilitate infiltration of endogenous NK cells and initiate potent antitumor immune responses. These observations are in good agreement with a previous report showing that activated NK cells secrete IFN-γ, TNF-α, and granulocyte–macrophage colony-stimulating factor to efficiently inhibit tumor growth [17].

## 4. Materials and Methods

### 4.1. Ethics Statement

All study samples were obtained following acquisition of the study participants’ written informed consent, in accordance with the Declaration of Helsinki. This research protocol was reviewed and approved by the institutional review board of Seoul National University Hospital (Permit Number: H-1004-027-315; date of approval: 1st June 2010). 

Mice were maintained in a laminar air flow cabinet with specific pathogen-free conditions. All facilities were approved by the Association for Assessment and Accreditation of Laboratory Animal Care. The number of animals used was minimized, and all necessary precautions were taken to alleviate pain or suffering. All animal studies were conducted according to the institutional guidelines established by the Hanyang University Institutional Animal Care and Use Committee (Approval Number: 2017-0187A; Date: 20th October 2017).

### 4.2. Ex Vivo Expansion and Cryopreservation of NK Cells

*Ex vivo* expansion of NK cells from healthy donor PBMCs was performed by GC LabCell (Yongin, Korea) as described previously [44]. Briefly, CD3^+^-depleted cells were seeded in CellGro SCGM medium (CellGenix, Freiburg, Germany) with 1–2% donor-plasma, γ-irradiated (2000 rad, ^137^Cs source) feeder PBMCs (5 × 10^7^ cells), 500 IU/mL IL-2 (Norvartis, Basel, Switzerland), and 10 ng/mL anti-CD3 monoclonal Ab OKT3 (eBioscience, San Diego, CA, USA) in an A-350N culture bag (NIPRO, Osaka, Japan). Cultured NK cells were restimulated on day 7 of the culture by adding irradiated feeder PBMCs. OKT3 was added to culture medium to preserve activating function of irradiated PBMCs to aid NK cell expansion. Fresh medium with 500 IU/mL IL-2 was fed into NK cell cultures every two days to maintain a concentration at 0.5–2 × 10^6^ cells/mL for 3-week culture. NK cells were thawed in water bath (37 °C) and used immediately for experiments without recovery period [90]. Cell count and viability were assessed by propidium iodide (PI) staining using an automatic cell counter (Digital Bio, Seoul, Republic of Korea). 

### 4.3. Cell Culture

Human pancreatic carcinoma cell lines (MIA PaCa-2, PANC-1, AsPC-1, and Capan-1) and human leukemia cell line (K562) were purchased from the American Type Culture Collection (Manassas, VA, USA). Cells were cultured in high glucose Dulbecco’s Modified Eagle’s Media (PAN Biotech, Dorset, UK) containing 10% fetal bovine serum (FBS; Alpha bioregen, Boston, MA, USA) in an incubator at 37 °C with 5% CO_2_. 

### 4.4. Immunostaining and Flow Cytometric Analysis

The expanded NK cells were stained with the following monoclonal Abs; anti-CD3- fluorescein isothiocyanate (FITC), anti-CD14-FITC, anti-CD16- phycoerythrin (PE), anti-CD19-PE, anti-DNAM-1-PE, anti-CD56-PE-cyanine (Cy)5, anti-CXCR3-PE, anti-NKp30-PE, anti-NKp44-PE, anti-NKp46-PE (all from BD Biosciences, San Jose, CA, USA), anti-NKG2A-PE, anti-NKG2C-PE, and anti-NKG2D-PE (all from R&D systems, Minneapolis, MN, USA). Stained NK cells were acquired on LSR Fortessa and data were analyzed using FlowJo software (TreeStar Inc., Ashland, OR, USA). To confirm phenotypic changes in NK cell receptors in co-culture of tumor cells with NK cells, cryopreserved NK cells were thawed and co-cultured with MIA PaCa-2 at an E:T ratio of 1:1. On 1, 2, or 3 days after co-culture, the harvested NK cells were immunostained with anti-CD16-PE, anti-NKp30-PE, anti-NKp44-PE, anti-NKp46-PE, anti-DNAM-1-PE, anti-CXCR3-PE (all from BD Biosciences, San Jose, CA, USA), anti-NKG2D-PE (R&D systems, Minneapolis, MN, USA), anti-CD96-PE, and anti-CD161-PE (all from eBioscience, San Diego, CA, USA) then analyzed by flow cytometry, as described above.

### 4.5. Intracellular Cytokines and CD107a Staining

To measure the expression levels of intracellular cytokines and CD107a by NK cells, NK cells were co-cultured with tumor targets at an E:T ratio of 1:1 for 4 h in the presence of anti-CD107a-allophycocyanin (APC) (BD Biosciences), monensin (GolgiStop; BD Biosciences, Eugene, OR, USA), and brefeldin A (GolgiPlug; BD Biosciences). After 4 h, cells were washed with fluorescence-activated cell sorter (FACS) flow buffer (BD Biosciences) and stained with anti-CD3-FITC (BD Biosciences), anti-CD56-APC-eFluor®780 (BD Biosciences), and 7-aminoactinomycin (BD Biosciences). Subsequently, the samples were permeabilized by BD CytoFix/CytoPerm^TM^ (BD Biosciences) and stained with anti-IFN-γ-PE (BD Biosciences) or anti-TNF-α-PE-Cy7 (eBioscience). Stained cells were acquired on an LSR Fortessa and data was analyzed using FlowJo software (TreeStar Inc.).

### 4.6. Calcein-AM Release Cytotoxicity Assay

Cytotoxicity of NK cells against tumor target cell lines was assessed by fluorometric cytotoxicity assay. Tumor cells were stained with 30 mM calcein-acetoxymethyl (AM) (Molecular probe, Eugene, OR, USA) for 1 h at 37 °C. NK cells were prepared at the E:T ratio of 10:1 per 1 × 10^4^ target cells. NK cells were co-cultured with labeled human pancreatic carcinoma cell lines (AsPC-1, MIA PaCa-2, Capan-1, and PANC-1) or a human leukemia cell line (K562) in 96-well plate in triplicate at corresponding ratios at 37 °C and 5% CO_2_ for 4 h under light protection. RPMI1640 containing 10% FBS or 0.1% triton-X100 was added to the targets to provide spontaneous and maximum release. The measurement was conducted at excitation 485 nm and emission 535 nm with the fluorometer. The percentage of specific calcein-AM release was calculated according to the formula % specific release = ((mean experimental release−mean spontaneous release)/(mean maximal release−mean spontaneous release)) × 100.

For blocking experiments, NK cells were preincubated with 20 μg/mL of anti-mouse IgG (BD Biosciences, San Jose, CA, USA), 10 μg/mL of anti-DNAM-1 (BD Biosciences, San Jose, CA, USA), 2 μg/mL of anti-NKG2D (R&D systems, Minneapolis, MN, USA), 10 μg/mL of anti-NKp44 (Biolegend, San Diego, CA, USA), or 2 μg/mL of anti-NKp30 (Biolegend) Abs for 30 min at 4 °C. The preincubated NK cells were co-incubated with MIA PaCa-2, AsPC-1, or Capan-1 cells at an E:T ratio 30:1 then the cytotoxicity of NK cells were assessed by calcein-AM release assay. The inhibition of cytotoxicity was calculated as a percentage of the inhibition by the isotype control Ab (anti-mouse IgG Ab).

### 4.7. MTT Assay

To evaluate the cancer cell-killing effect of human NK cells, MIA PaCa-2 and PANC-1 cells were grown to 50–60% confluence in 96-well plates and incubated with human NK cells at various E:T ratios. At 48, 72, and 96 h post incubation, 150 μL of MTT (Sigma-Aldrich, St. Louis, MO, USA) at 2 mg/mL in PBS was added to each well and incubated at 37 °C for 4 h. The supernatant was then removed, and the precipitate was dissolved in 200 μL of dimethyl sulfoxide (DMSO). Plates were read on a microplate reader at 540 nm. The absorbance obtained from cells in a PBS-treated group was considered as 100% cell viability.

### 4.8. Apoptosis Analysis by Flow Cytometer

Induction of apoptosis was analyzed with Annexin V-FITC/PI apoptosis detection kit (BD Biosciences, San Jose, CA, USA) according to the manufacturer’s instructions. MIA PaCa-2 and PANC-1 cells were seeded in 6-well plates and incubated with human NK cells at various E:T ratios. Treated cells were washed twice with cold PBS and resuspended in 1× binding buffer at a concentration of 10^6^ cells per mL. Cells were mixed with 5 μL of FITC-conjugated annexin-V reagent and 5 μL of PI. After 15 min incubation at room temperature in the dark and further washings, the samples were analyzed by FACSCalibur analyzer (BD Biosciences, San Jose, CA, USA) with Cell Quest software (BD Biosciences, San Jose, CA, USA). Approximately 10,000 events were counted for each sample. 

### 4.9. In Vivo Antitumor Efficacy and Bioluminescence Imaging

The orthotopic pancreatic cancer model was established by injecting 5 × 10^6^ firefly luciferase-expressing MIA PaCa-2 cells into the pancreas of 5-week-old male nude mice (Charles River Korea Inc., Seoul, Korea). After 14 days postimplantation (designated as day 0), bioluminescence imaging was taken to confirm the establishment of the orthotopic pancreatic tumor model. Mice were randomized into four groups (PBS, gemcitabine, IL-2, or NK cells (n = 8, each group) to receive intravenously administered human NK cells (2 × 10^7^ cells) 3 times a week for 2 weeks or intraperitoneally administered gemcitabine (100 mg/kg) twice a week for 2 weeks. To promote NK cells activity *in vivo*, mice received an intraperitoneal injection of IL-2 (250 U) on the day of NK cell injections and 2 times daily for 2 weeks. Tumor growth was measured every 7 days after the first treatment by bioluminescence imaging using the IVIS imaging system (Xenogen, Alameda, CA, USA). *In vivo* bioluminescence signals were calculated as the sum of both prone and supine acquisitions for each mouse after background subtraction of total flux (photons/sec (p/s)) from a total body region of interest.

### 4.10. Quantification of IFN-γ and TGF-β Expression

Tumor tissues and serum were harvested from mice treated with NK cells at 3 days after last treatment. Tissues were homogenized and liquefied in PBS containing protease inhibitor cocktail (Sigma-Aldrich, St. Louis, MO, USA). IFN-γ and TGF-β in tumor tissue extracts or serums were measured by conventional enzyme-linked immunosorbent assay (ELISA) kits (IFN-γ and TGF-β ELISA kit: R&D Systems, Minneapolis, MN, USA). Results were normalized to total protein concentration per tumor and calculated as picograms per milligram of total protein.

### 4.11. Histological and Immunohistochemical Analysis

Tumor tissues were collected from mice at 3 days after the final treatment. Tumor tissues were fixed in 10% formalin, processed for paraffin embedding, and cut into 5 μm sections. Representative tissue slides were stained with H & E and examined by microscopy. Tumor sections were immunostained with PCNA-specific Ab (Dako, Glostrup, Denmark) to determine the effects on tumor cell proliferation. The TUNEL assay was performed according to manufacturer’s instructions (Merck, Darmstadt, Germany) to detect apoptotic cell population following treatment. To detect human and murine NK cells, tumor tissues were stained with mouse anti-human CD56 Ab (R&D Systems, Minneapolis, MN, USA) or rat anti-mouse DX5 Ab (Biolegend). After incubation with primary Ab at 4 °C overnight, the sections were incubated with biotinylated goat anti-mouse Ab (BD Biosciences) or biotinylated goat anti-rat Ab (BD Biosciences, San Jose, CA, USA). The sections were further incubated with streptavidin horseradish peroxidase solution (BD Biosciences, San Jose, CA, USA). Nuclear staining with 4,6-diamidino-2-phenylindole (Sigma-Aldrich, St. Louis, MO, USA) was also performed. All slides were counterstained with Meyer’s hematoxylin (Sigma-Aldrich, St. Louis, MO, USA).

### 4.12. Assessment of NK Cell Biodistribution 

The orthotopic pancreatic tumor-bearing mice were injected intravenously with NK cells (2 × 10^7^) three times a week for 2 weeks or with PBS as a control. The spleen, liver, stomach, heart, lung, pancreas, tumor, muscle, and kidney tissues were harvested at 1, 2, or 7 days after the final injection. The tissues were fixed in 10% formalin, embedded in paraffin, and cut into 5-μm-thick sections. The NK cells in each sample were assessed by immunohistochemistry as described above.

### 4.13. Statistical Analysis

Data were expressed as mean ± standard deviation (SD). Statistical significance was determined by two-tailed Student’s *t*-test (SPSS 13.0 software; SPSS, Chicago, IL, USA). Data with *p*-values less than 0.05 were considered statistically significant. 

## 5. Conclusions

Our findings show that large-scale expanded and cryopreserved NK cells derived from KIR haplotype B healthy donors can efficiently home to and infiltrate into desmoplastic tumor tissues. The intratumoral infiltration of allogeneic human NK cells led to high expression levels of antitumor cytokine (IFN-γ) and attenuated the expression levels of immunosuppressive TGF-β, ultimately alleviating tumor-induced immunosuppression. Additionally, allogeneic human NK cells effectively induced apoptosis of tumor cells to elicit potent antitumor effects. These results demonstrate that allogeneic NK cells are a promising candidate to induce potent therapeutic antitumor activity. 

## Figures and Tables

**Figure 1 cancers-11-00966-f001:**
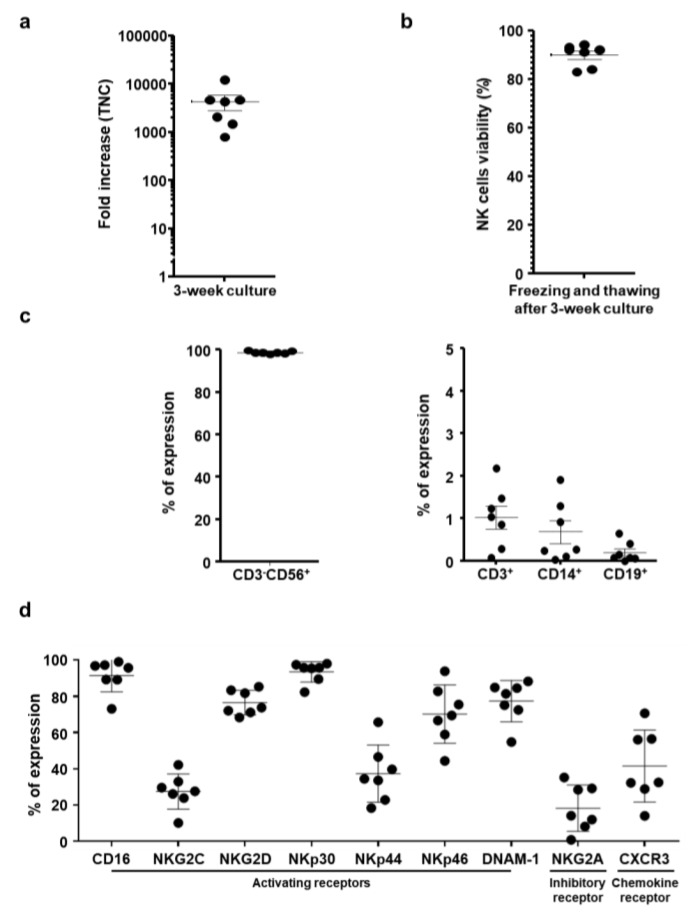
Characteristics of expanded and frozen natural killer (NK) cells. (**a**) The fold change in total nucleated cell count (TNC) of *ex vivo*-expanded NK cells between day 0 and day 21 of expansion. (**b**) Cell viability of cryopreserved NK cells after thawing. The viability of NK cells was evaluated by staining with propidium iodide to detect dead cells. % viable cells = number of viable cells/initial population frozen. (**c**) Purity of NK cells after thawing. The percent of cluster of differentiation (CD)3^−^CD56^+^, CD3^+^, CD14^+^, and CD19^+^ cells were analyzed by flow cytometry. (**d**) Surface expression of activating receptors, inhibitory receptors, and chemokine receptors was analyzed by flow cytometry after NK cell expansion. Results indicate the percentage of reactive NK cells within each NK subset (n = 7). Data are representative of two independent experiments. Each dot represents the results from one individual.

**Figure 2 cancers-11-00966-f002:**
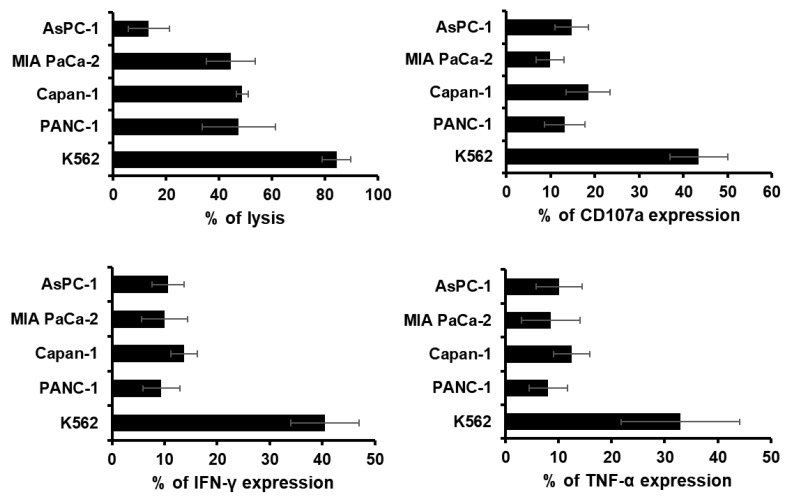
Cytotoxic effect of cryopreserved natural killer (NK) cells following co-culture with cancer cells. Direct cytolytic activity of NK cells was assessed by calcein-acetoxymethyl (AM) release assay using K562 and various pancreatic cell lines (AsPC-1, MIA PaCa-2, Capan-1, and PANC-1) at an effector-to-target (E:T) ratio of 10:1. The levels of degranulation marker cluster of differentiation (CD)107a and cytokine (interferon (IFN)-γ and tumor necrosis factor (TNF)-α) expression on NK cells were assessed by flow cytometry following co-incubation of K562 or pancreatic cancer cells with NK cells at an E:T ratio of 1:1. Data are representative of three independent experiments. Results show the mean percentages ± standard deviation (SD) of reactive NK cells within each NK subset (n = 7).

**Figure 3 cancers-11-00966-f003:**
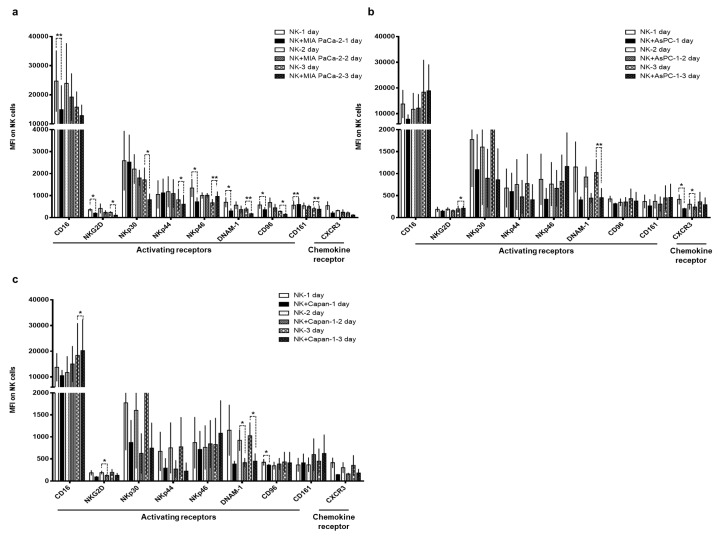
Phenotypic changes in natural killer (NK) cell receptors following co-culturing with cancer cells. NK cells were co-incubated with various pancreatic cell lines (MIA PaCa-2, AsPC-1, and Capan-1) at an effector-to-target (E:T) ratio of 1:1 in the presence of human interleukin (IL)-2. NK cells were collected, and the expression levels of various NK cell receptors were analyzed by flow cytometry on day 1, 2, and 3 of co-incubation. The data are representatives of three independent experiments performed in triplicate (**a**–**c**). Data are shown as mean ± standard deviation (SD); * *p* < 0.05, ** *p* < 0.01.

**Figure 4 cancers-11-00966-f004:**
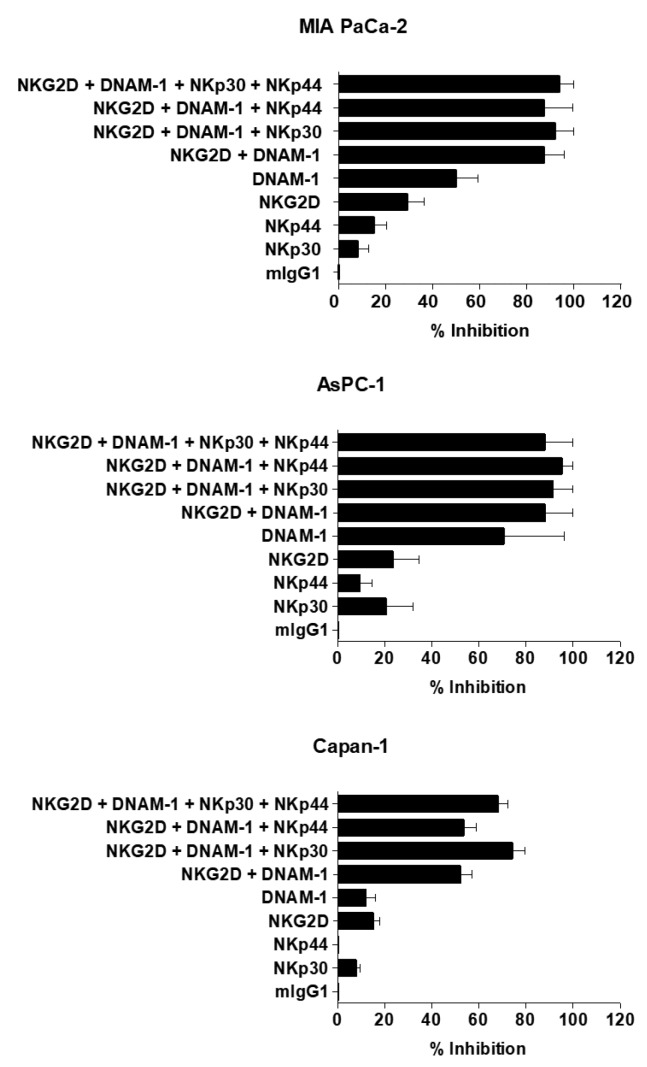
Inhibition of natural killer (NK) cell-mediated cytocidal effect against pancreatic cancer cells by blocking of various NK cell activating receptors. NK cells were preincubated with a single or combination of several blocking antibodies targeting NKp30, NKp44, NKG2D and/or DNAM-1. Then, the NK cells were co-cultured with MIA PaCa-2, AsPC-1, or Capan-1 at an effector-to-target (E:T) ratio of 30:1 for 4 h. The cytotoxicity was analyzed by calcein-acetoxymethyl (AM) release assay. The inhibition of cytotoxicity was calculated as a percentage of the inhibition by the isotype control antibody. The assay was performed two times with expanded NK cells from different donors, and representative data are presented. Results indicate the mean percentages ± standard deviation (SD) of reactive NK cells within each NK subset (n = 4).

**Figure 5 cancers-11-00966-f005:**
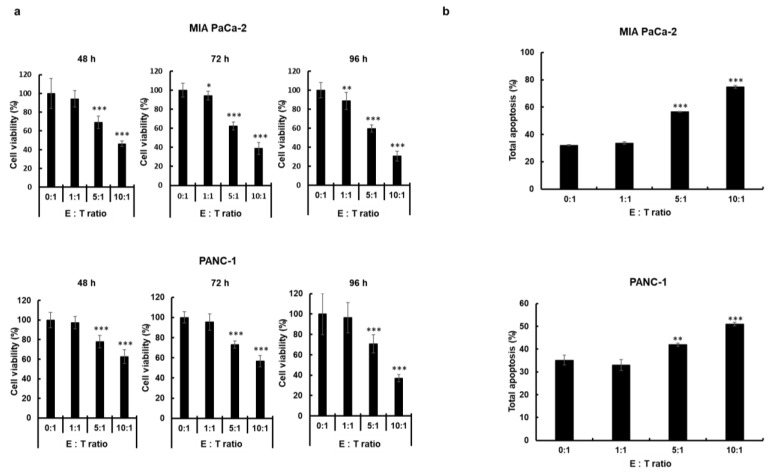
Pancreatic cancer cell killing effect of the natural killer (NK) cells via induction of apoptosis. (**a**) Human pancreatic cancer cells (MIA PaCa-2 and PANC-1) were incubated with NK cells at various effector-to-target (E:T) ratios. At 48, 72, and 96 h after treatment, cell viability was measured by 3-(4,5-dimethylthiazol-2-yl)-2,5-diphenyltetrazolium bromide (MTT) assay. Each cell line was tested at least three times, and data are shown as mean ± standard deviation (SD) of triplicate experiments; * *p* < 0.05, ** *p* < 0.01, *** *p* < 0.001. (**b**) Human pancreatic cancer cells (MIA PaCa-2 and PANC-1) were incubated with NK cells at various E:T ratios (0:1, 1:1, 5:1, and 10:1). At 24 h after treatment, apoptosis was measured by Annexin V-fluorescence isothiocyanate (FITC)/propidium iodide (PI) double-staining. The data are representatives of three independent experiments performed in triplicate. Data presented mean ± SD; ** *p* < 0.01, *** *p* < 0.001.

**Figure 6 cancers-11-00966-f006:**
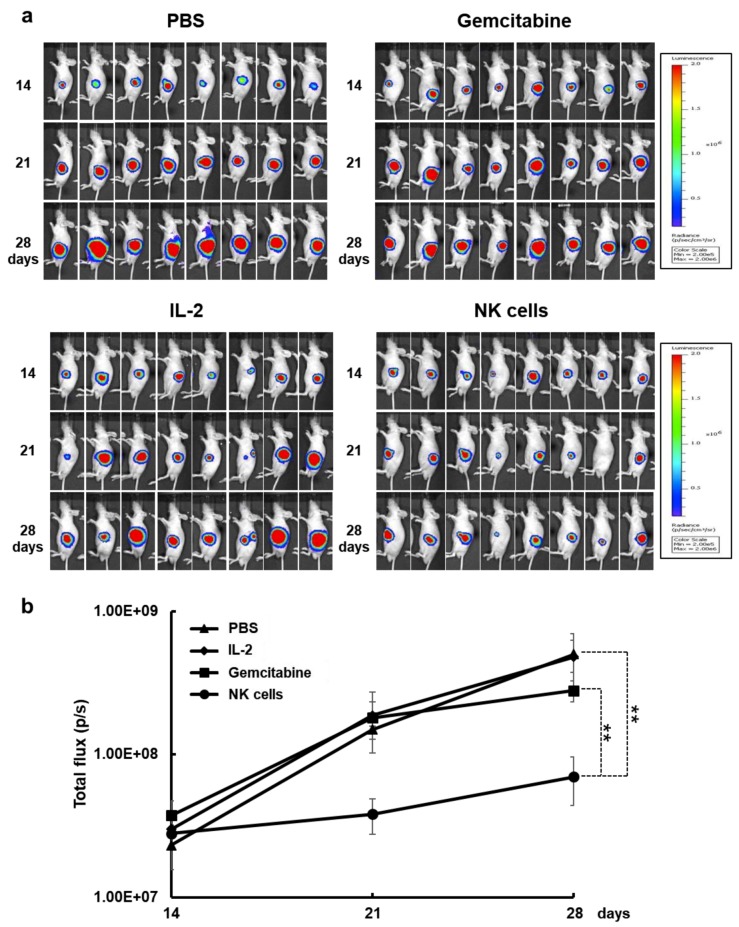
Therapeutic efficacy of natural killer (NK) cells in a MIA PaCa-2 orthotopic pancreatic tumor model. Orthotopic pancreatic tumor models were established by injecting firefly luciferase-expressing MIA PaCa-2 cells (5 × 10^6^) into the pancreas of nude mice. At 14 days after tumor cell injection, 2 × 10^7^ NK cells three times a week for 2 weeks or gemcitabine at 100 mg/kg twice a week for 2 weeks was intraperitoneally administered. (**a**) Bioluminescence whole body imaging was monitored every 7 days following the treatments. (**b**) Bioluminescence signals were calculated after background subtraction in total flux photons/sec from a body region of interest. These images are representatives from three independent experiments. Data presented as mean ± standard deviation (SD); ** *p* < 0.01.

**Figure 7 cancers-11-00966-f007:**
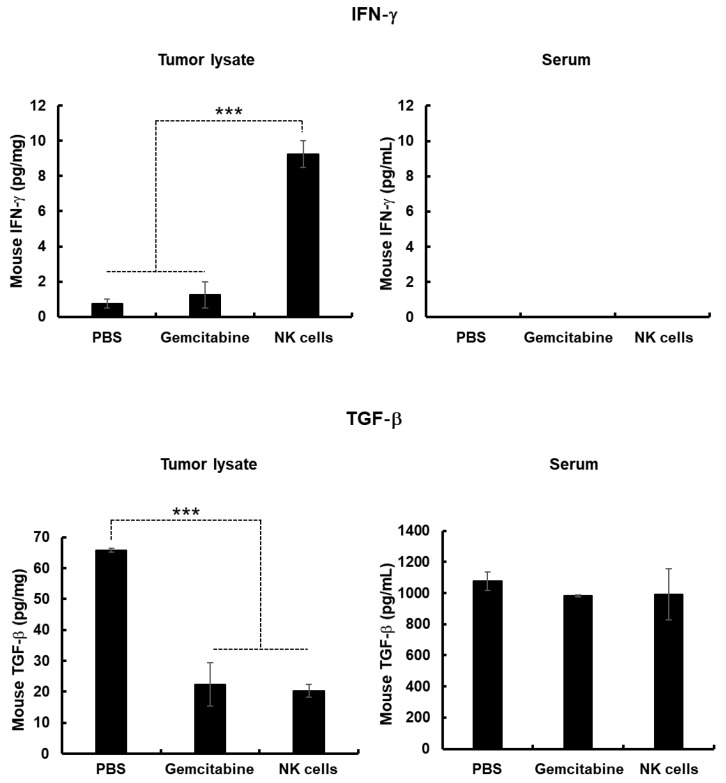
Expression of interferon (IFN)-γ and transforming growth factor (TGF)-β in tumor tissues and serum. Tumor tissues and serum were obtained at 3 days after final treatment with phosphate-buffered saline (PBS), gemcitabine, or natural killer (NK) cell for quantitation of IFN-γ and TGF-β by conventional enzyme-linked immunosorbent assay (ELISA) kit. Data are presented as mean expression level ± standard deviation (SD) of triplicate experiments with at least three mice per group. *** *p* < 0.001.

**Figure 8 cancers-11-00966-f008:**
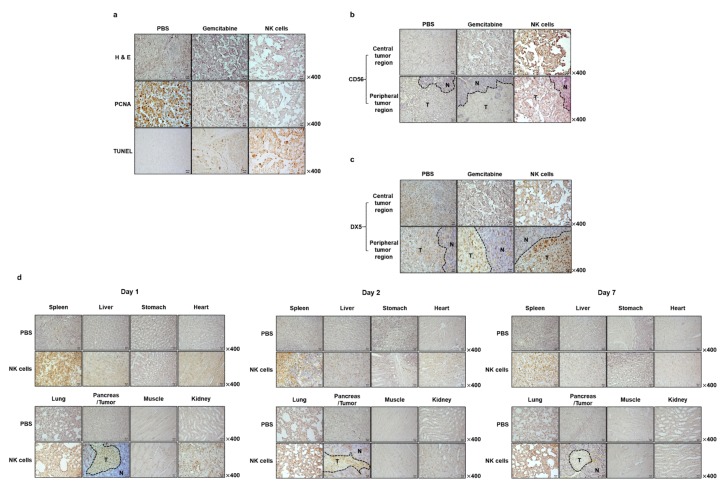
Histological and immunohistochemical analyses of tumor tissues from mice treated with natural killer (NK) cells. Tumor tissues were collected from mice at 3 days after the final treatment. (**a**) Representative sections were stained with hematoxylin and eosin (H & E), and the expression of proliferating cell nuclear antigen (PCNA) was assessed by immunohistochemistry. Terminal deoxynucleotidyl transferase dUTP nick end labeling (TUNEL) assays were performed to assess induction of apoptosis in tumor tissues. Tumor tissues were stained with (**b**) anti-CD56 or (**c**) anti-DX5 antibody to assess intratumoral infiltration of NK cells. Original magnification, ×400. Images are representatives of results from three independent experiments. (**d**) NK cells were systemically administered 3 times a week for 2 weeks to MIA PaCa-2 orthotopic tumor-bearing mice. Spleen, liver, stomach, heart, lung, pancreas, tumor, muscle, and kidney tissues were harvested at 1, 2, or 7 days post-final injection to evaluate the biodistribution profiles of NK cells. Immunohistochemistry was performed to analyze human NK cell accumulation in each tissue. Original magnification, ×400. Similar results were obtained from at least three separate experiments. T, tumor tissues; N, normal tissues.

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
