# Peer review of "Cryopreserved Human Natural Killer Cells Exhibit Potent Antitumor Efficacy against Orthotopic Pancreatic Cancer through Efficient Tumor-Homing and Cytolytic Ability"

_cancers, 2019, doi:10.3390/cancers11070966_

Round 1
Reviewer 1 Report
In this article, the authors describe the phenotype and biological function of human NK cells after freezing and thawing and propose them as a feasible off-the-shelf option for NK-mediated anticancer immunotherapy.
The work is well-organised and the results are clearly described and discussed.
My major comment is that a direct comparison with freshly isolated NK cells in all the experiments, especially the functional ones, would be important to demonstrate that thawed NK cells are not significantly less effective. This is particularly important if they want to propose thawed NK cells as an alternative to freshly isolated ones, because it would not be ethical to use a therapy that is not as effective as the best option.
I strongly recommend to have freshly isolated NK cells in all their experiments as a reference.
One additional point is that xenografts in nude mice don't usually generate a desmoplastic stroma of similar extent to what observed in patients, therefore their conclusion about the homing ability of NK cells cannot really be drawn using this experimental method.
Minor points:
Figure 2: they mention a E:T ratio of 1:1, however, in the method section the ratio used for the calcein-AM release assay is 10:1. Furthermore, if a ratio 1:1 induces about 50% of cell lysis in 4h, then I find it hard to explain why in figure 4 they only have an effect of similar extent with a 10:1 ratio and after 48h. Please double-check the ratio used in figure 2.
Figure 3: if they would represent the data as ratio NK+tumor cells/NK only, they could have one single graph for each cell line and the figure would be smaller and easier to read.
Methods: the mouse experiments should also be mentioned in the ethics statements.
Author Response
June 12, 2019
Professor Danica Jevtic
Assistant Editor, Cancers
Re: Cryopreserved human natural killer cells exhibit potent antitumor efficacy against orthotopic pancreatic cancer through efficient tumor homing and cytolytic ability
Dear Dr. Danica Jevtic,
Thank you for accepting our paper for revision. We fully appreciate the thoughtful reviewer comments as suggested are submitting a revised version of our manuscript, which addresses specific issues brought forth by the reviewer. Detail description of our response and changes are outlined below.
We have noticed that there is no Conclusions part in the manuscript. Please add it during revisions.
A) As suggested by the editor, we have created a Conclusion section on page 27 in our revised manuscript and have utilized the last part of the discussion as following; “Conclusion. Our findings show that large-scale expanded and cryopreserved NK cells derived from KIR haplotype B healthy donors can efficiently home to and infiltrate into desmoplastic tumor tissues. The intratumoral infiltration of allogeneic human NK cells led to high expression levels of antitumor cytokine (IFN-g) and attenuated the expression levels of immunosuppressive TGF-b, ultimately alleviating tumor-induced immunosuppression. Additionally, allogeneic human NK cells effectively induced apoptosis of tumor cells to elicit potent antitumor effects. These results demonstrate that allogeneic NK cells are a promising candidate to induce potent therapeutic antitumor activity.” Thank you.
Reviewer 1:
In this article, the authors describe the phenotype and biological function of human NK cells after freezing and thawing and propose them as a feasible off-the-shelf option for NK-mediated anticancer immunotherapy. The work is well-organised and the results are clearly described and discussed.
Q1) My major comment is that a direct comparison with freshly isolated NK cells in all the experiments, especially the functional ones, would be important to demonstrate that thawed NK cells are not significantly less effective. This is particularly important if they want to propose thawed NK cells as an alternative to freshly isolated ones, because it would not be ethical to use a therapy that is not as effective as the best option. I strongly recommend to have freshly isolated NK cells in all their experiments as a reference.
A1) In depth comparison of fresh and post-thawing NK cells has been performed in a previous report by GreenCross LabCell (Immune Netw. 2018 Aug;18(4):e31). Their findings show that cell recovery rate was similar for both cryopreserved and freshly expanded NK cells, as shown in the figure below. Similarly, cytotoxic activities against K562 cells and various hepatocellular carcinoma cells were retained at a similar level both before and after cryopreservation. In addition, the cryopreservation of expanded NK cells did not impair the degranulation activity measured by CD107a expression and the secretion of IFN-g and TNF-a. Only notable difference between freshly isolated and cryopreserved NK cells was slightly reduced NK cell viability in cryopreserved samples.
This previous report also analyzed the effect of cryopreservation on receptor expression profile of NK cells, as presented below. Cryopreserved in the figure below, which denotes after freezing and thawing of 21-day expansion, is same condition as NK cells utilized in our current study (Immune Netw. 2018 Aug;18(4):e31). Day 0 on the graph means right before ex vivo expansion. As shown below, NK cell surface profile (activating receptors, inhibitory receptors, and chemokine receptors on NK cells) from day 21 of expansion was not significantly altered by freezing and thawing, with an exception of NKp46.
Based on these background, we firmly believe that improvement in off-the-shelf availability through cryopreservation of allogeneic NK cell banking would greatly expedite the treatment process by circumventing on-site preparation.
Q2) One additional point is that xenografts in nude mice don't usually generate a desmoplastic stroma of similar extent to what observed in patients, therefore their conclusion about the homing ability of NK cells cannot really be drawn using this experimental method.
A2) Masson’s trichrome stained orthotopic pancreatic xenograft tumor and pancreatic cancer patient-derived tumor spheroids from our previous reports have been presented below (Int J Cancer. 2018 Jan 15;142(2):392-413; Cancer Lett. 2019 May 28; Epub ahead of print). As seen in this Figure, orthotopic human pancreatic xenograft tumors (established using same methodology and cell line as our present study) closely emulated the desmoplastic stroma observed in pancreatic cancer patient-derived xenograft (PDX) tumor tissues.
Q3) Figure 2: they mention a E:T ratio of 1:1, however, in the method section the ratio used for the calcein-AM release assay is 10:1. Furthermore, if a ratio 1:1 induces about 50% of cell lysis in 4h, then I find it hard to explain why in figure 4 they only have an effect of similar extent with a 10:1 ratio and after 48h. Please double-check the ratio used in figure 2.
A3) We appreciate Reviewer’s valuable comment. We realized how interpretation of our data from Figure 2 in Results section on page 8 might have been misleading, as we did not explain that E:T ratio was different for cytotoxicity assay (E:T ratio = 10:1) and our assessment of degranulation marker/cytokines (E:T ratio = 1:1). We have removed misleading E:T ratio mentioned within Results section and the E:T ratios for both assays are better represented in revised Figure Legend on page 9 as following; “…calcein-AM release assay using K562 and various pancreatic cell lines (AsPC-1, MIA PaCa-2, Capan-1, and PANC-1) at an effector:target (E:T) ratio of 10:1. The levels of degranulation marker CD107a and cytokine (IFN-γ and TNF-α) expression on NK cells were assessed by flow cytometry following co-incubation of K562 or pancreatic cancer cells with NK cells at an E:T ratio of 1:1.”
Reviewer commented on the similar anticancer effect by NK cells at two different incubation period of 4 and 48 hr post treatment for calcein-release assay (Figure 2) and MTT (Figure 4), respectively. This is likely due to calcein-release assay not accommodating for rapid rate of cancer cell proliferation due to brief co-culture period. In vitro evaluation of the lytic activity of NK and T cells via calcein-release assay has been used as a standardized and routine method of evaluating cytotoxicity (CLINICAL AND DIAGNOSTIC LABORATORY IMMUNOLOGY. 2001 Nov;8(6):1131-1135). In MTT assay, formazan is created from cancer cells, and viability of cellular populations are obtained. Together, these factors would mean that rapid proliferation rate of cancer cells may lead to underestimation of overall cytolytic effect that NK cell against cancer cells in Figure 4.
Q4) Figure 3: if they would represent the data as ratio NK+tumor cells/NK only, they could have one single graph for each cell line and the figure would be smaller and easier to read.
A4) Upon the Reviewer’s request, Figure 3 has been reformatted to have each cell line results be represented as a single graph.
The following is Figure 3
Q5) Methods: the mouse experiments should also be mentioned in the ethics statements.
A5) As suggested by the reviewer, we have added the ethics statement to the Materials and Methods section on page 28 as following; “Mice were maintained in a laminar air-flow cabinet with specific pathogen-free conditions. All facilities were approved by the Association for Assessment and Accreditation of Laboratory Animal Care. The number of animals used was minimized, and all necessary precautions were taken to alleviate pain or suffering. All animal studies were conducted according to the institutional guidelines established by the Hanyang University Institutional Animal Care and Use Committee.”
The newly revised manuscript also follows the editorial instructions regarding style and format. With these responses and changes, we hope the manuscript is now acceptable for publication in Cancers. Please feel free to contact me if you have any questions.
Sincerely yours,
Chae-Ok Yun, Ph.D.
Professor
Department of Bioengineering
College of Engineering
Hanyang University
17 Haengdang-Dong, Seongdong-Gu, Seoul, Korea
Tel: 82-2-2220-0491
Fax: 82-2-2220-4850
E-mail: chaeok@hanyang.ac.kr

Reviewer 2 Report
The review by Eonju Oh et al. analyze the cytotoxic activity of allogeneic in vitro expanded NK cells which have been cryopreserved and thawed. They use pancreatic cancer cells for their experiments. The paper is well-written and have extensive data. However, there are some issues which are not clear and require attention.
Major changes or questions:
1. For all the experiments performed authors have used cryopreserved and thawed cells. However, authors do not specify whether NK cells have been left overnight or a few hours to recover of the cryopreservation and thawing. Are NK cells used straight away?
2. Authors have not performed any experiment comparing the efficacy of fresh NK cells and after thawing. This should be done to confirm that cryopreservation does not decrease NK cell activity.
Results:
Page 3, Line 102: how is it possible to expand NK cells with OKT3 if NK cells do not express CD3?
Figure 1D: It is relevant to know the change of expression of these receptors after thawing. Do you have the comparison of expression before and after thawing?
Page 5, line 165: Authors state: “These results suggest that there are significant alterations to the surface expression levels of various activating receptors of NK cells during the early stages of cell-to-cell contact..”. I don’t think early stages refer to a period from 24-36h. Either authors analyze them at early stages (less than 1h) or do not refer early stages as a period from 24-36h.
Figure 3: What is the killing of these assays for each cell line after 36h? Were tumors cells eliminated? Also, for NK+MIA PaCa-2, NKP44 was increased after 36h co-culture and authors state that it decreases.
Page 9: 2.3. Cytotoxic Effect of NK Cells Against Human Pancreatic Cancer Cell Lines: Authors indicate that they perform an MTT assay to look at cytotoxicity from 48 to 96h.
MTT is a cell proliferation assay and cannot be used as a cytotoxic assay, because you can’t know which cell population is being quantified unless you label both cell populations with specific markers. It might occur that the tumor cells proliferate, especially at 96h, when NK cells have not been able to eliminate all tumor cells. Same issue applies for the quantification of annexinV-PI. Have the authors any marker to differentiate between both cell populations?
Figure 6: authors state that NK cells exhibited potent antitumor activity. However, they inject a very high dose of NK cells: 20x106 three times a week for 2 weeks, and even with this dose NK cells do not achieve complete abrogation of the disease. Therefore, even though NK cells are better than gemcitabine, it does not look as a very potent antitumor activity. May be the sentence potent antitumor activity should be modified.
Figure 7: authors quantify mouse IFNg. However, IFNg should be mainly produced by human NK cells. Did authors look at human IFNg?
Minor changes:
Abstract:
Define ECM
Line 25: “such as poor availability in a readily usable form due to arduous production”: it is not clear the meaning of the sentence.
Page 2, Line 88-89: “These NK cells induced potent cytotoxic effects against human pancreatic cancer cells in vitro…”. Clarify whether these NK cells are thawed NK cells after cryopreservation.
Page 16, Line 333: authors state: “NK cells were also detected in the lungs at all time points, suggesting that systemically administered NK cells can be trapped in the lungs as a “first pass” tissue after systemic injection”. Trapping of cells in the lungs occurs always in mice after i.v. injections of cells, not only for NK cells. Authors should delete that statement.
Page 4, line 144: after TNFa, it appears a strange symbol instead of a brake.
Figure 2: In the text states that cytotoxicity assays were performed at 1:1 E:T ratio, and in the figure legend it states 10:1 E:T ratio. Which one is it?
Discussion:
Line 381: “Interestingly, these NK cells showed lower expression level of activating receptors after co-cultivation with pancreatic cancer cells”. This phenomenon has already been described as trogocytosis. Authors should include it in the discussion.
Figure 3, 5 and 8 are divided in different pages. They need to be edited to fit it into one page each figure.
Author Response
June 12, 2019
Professor Danica Jevtic
Assistant Editor, Cancers
Re: Cryopreserved human natural killer cells exhibit potent antitumor efficacy against orthotopic pancreatic cancer through efficient tumor homing and cytolytic ability
Dear Dr. Danica Jevtic,
Thank you for accepting our paper for revision. We fully appreciate the thoughtful reviewer comments as suggested are submitting a revised version of our manuscript, which addresses specific issues brought forth by the reviewer. Detail description of our response and changes are outlined below.
We have noticed that there is no Conclusions part in the manuscript. Please add it during revisions.
A) As suggested by the editor, we have created a Conclusion section on page 27 in our revised manuscript and have utilized the last part of the discussion as following; “Conclusion. Our findings show that large-scale expanded and cryopreserved NK cells derived from KIR haplotype B healthy donors can efficiently home to and infiltrate into desmoplastic tumor tissues. The intratumoral infiltration of allogeneic human NK cells led to high expression levels of antitumor cytokine (IFN-g) and attenuated the expression levels of immunosuppressive TGF-b, ultimately alleviating tumor-induced immunosuppression. Additionally, allogeneic human NK cells effectively induced apoptosis of tumor cells to elicit potent antitumor effects. These results demonstrate that allogeneic NK cells are a promising candidate to induce potent therapeutic antitumor activity.” Thank you.
Reviewer 2:
The review by Eonju Oh et al. analyze the cytotoxic activity of allogeneic in vitro expanded NK cells which have been cryopreserved and thawed. They use pancreatic cancer cells for their experiments. The paper is well-written and have extensive data. However, there are some issues which are not clear and require attention.
Q1) For all the experiments performed authors have used cryopreserved and thawed cells. However, authors do not specify whether NK cells have been left overnight or a few hours to recover of the cryopreservation and thawing. Are NK cells used straight away?
A1) We appreciate the Reviewer’s comment. Cryopreserved NK cells were used right after post-thawing without recovery period. We have clarified this issue in the Materials and Method section of our revised manuscript on page 28 as following; “NK cells were thawed in water bath (37°C) and used immediately for experiments without recovery period [90].”
Q2) Authors have not performed any experiment comparing the efficacy of fresh NK cells and after thawing. This should be done to confirm that cryopreservation does not decrease NK cell activity.
A2) In depth comparison of freshly isolated and post-thawing NK cells has been performed in previous report by GreenCross LabCell (Immune Netw. 2018 Aug;18(4):e31). Their findings show that cell recovery rate was similar for both cryopreserved and freshly expanded NK cells, as shown in the figure below. Similarly, cytotoxic activities against K562 cells and various hepatocellular carcinoma cells were retained at a similar level both before and after cryopreservation. In addition, the cryopreservation of expanded NK cells did not impair the degranulation activity measured by CD107a expression and the secretion of IFN-g and TNF-a. Only notable difference between freshly isolated and cryopreserved NK cells was slightly reduced NK cell viability in cryopreserved samples.
Based on these background, we firmly believe that improvement in off-the-shelf availability through cryopreservation of allogeneic NK cell banking would greatly expedite the treatment process by circumventing on-site preparation.
Q3) Page 3, Line 102: how is it possible to expand NK cells with OKT3 if NK cells do not express CD3?
A3) We appreciate the Reviewer’s comment. We have isolated allogeneic NK cells from healthy donor PBMCs and co-cultured them with irradiated feeder PBMCs to re-stimulate and expand NK cells. OKT3 was added to the NK cell culture to maintain activating function of irradiated feeder PBMCs. We have clarified this issue in the Materials and Method section of our revised manuscript on page 28 as following; “Cultured NK cells were re-stimulated on day 7 of the culture by adding irradiated feeder PBMCs. OKT3 was added to culture medium to preserve activating function of irradiated PBMCs to aid NK cell expansion.” Others have similarly reported usage of OKT3 and IL-2 in culture medium when NK cell expansion is performed ex vivo using irradiated PBMC as feeder cells (Front Immunol. 2017; 8: 854). For NK cell expansion, it is common for feeder cells to be irradiated and then supplemented with survival and activating factors like IL-2, IL-15, and OKT3 (J Immunother. 2013 Sep; 36(7):373-81; J Transl Med. 2015 Aug 25;13:277; Cancer Res. 2013 Apr 15;73(8):2608-2618). Activation of feeder cells enhances their indispensable cross-talk capacity with NK cells during expansion protocol, which maximizes ex vivo proliferation of NK cells.
Q4) Figure 1D: It is relevant to know the change of expression of these receptors after thawing. Do you have the comparison of expression before and after thawing?
A4) We appreciate reviewer’s valuable comment. The previous report by GreenCross LabCell (Immune Netw. 2018 Aug;18(4):e31) analyzed the effect of cryopreservation on receptor expression profile of NK cells, as presented below. Cryopreserved in the figure below, which denotes after freezing and thawing of 21-day expansion, is same condition as NK cells utilized in our current study. Day 0 on the graph means right before ex vivo expansion. As shown below, NK cell surface profile (activating receptors, inhibitory receptors, and chemokine receptors on NK cells) from day 21 of expansion was not significantly altered by freezing and thawing, with an exception of NKp46.
Q5) Page 5, line 165: Authors state: “These results suggest that there are significant alterations to the surface expression levels of various activating receptors of NK cells during the early stages of cell-to-cell contact..”. I don’t think early stages refer to a period from 24-36h. Either authors analyze them at early stages (less than 1h) or do not refer early stages as a period from 24-36h.
A5) We appreciate the Reviewer’s comment. The expression “early stages” has been removed in revised manuscript in Result section (page 10) as following; “These results suggest that there are significant alterations to the surface expression levels of various activating receptors of NK cells following cell-to-cell contact with pancreatic cancer cells.”
Q6) Figure 3: What is the killing of these assays for each cell line after 36h? Were tumors cells eliminated? Also, for NK+MIA PaCa-2, NKP44 was increased after 36h co-culture and authors state that it decreases.
A6) We have compared the surface expression levels of various activating receptors on NK cells cultured with or without pancreatic cancer cells (MIA PaCa-2, AsPC-1, and Capan-1) at 24 h intervals (1, 2, or 3 days of co-culture) by flow cytometry in Figure 3. As shown in Figure 3A, the expression of NKp44 was significantly reduced on 3 days after co-cultivation with MIA PaCa-2 compared to control NK cells.
Q7) Page 9: 2.3. Cytotoxic Effect of NK Cells Against Human Pancreatic Cancer Cell Lines: Authors indicate that they perform an MTT assay to look at cytotoxicity from 48 to 96h. MTT is a cell proliferation assay and cannot be used as a cytotoxic assay, because you can’t know which cell population is being quantified unless you label both cell populations with specific markers. It might occur that the tumor cells proliferate, especially at 96h, when NK cells have not been able to eliminate all tumor cells. Same issue applies for the quantification of annexinV-PI. Have the authors any marker to differentiate between both cell populations?
A7) As suggested by the Reviewer, we performed FACS analysis to differentiate between cancer and NK cell populations at 96 h after co-cultivation. Cancer and NK cells were differentiated based on their difference in size and granularity using flow cytometry. As shown below, flow cytometry-based analysis of cell killing effect, which could differentiate cancer and NK cells, showed potent cell killing efficacy in a dose-dependent manner as those achieved in Figure 5.
Q8) Figure 6: authors state that NK cells exhibited potent antitumor activity. However, they inject a very high dose of NK cells: 20x106 three times a week for 2 weeks, and even with this dose NK cells do not achieve complete abrogation of the disease. Therefore, even though NK cells are better than gemcitabine, it does not look as a very potent antitumor activity. May be the sentence potent antitumor activity should be modified.
A8) We agree with the Reviewer’s comment about antitumor activity. We have revised our manuscript in the Result section on page 15 as following; “As shown in Figure. 6A, NK cells elicited higher level of antitumor activity in respect to conventional chemotherapy option of gemcitabine.” Thank you.
Q9) Figure 7: authors quantify mouse IFNg. However, IFNg should be mainly produced by human NK cells. Did authors look at human IFNg?
A9) We agree with the Reviewer that it would have been beneficial to analyze human IFN-g expression level in the tumor tissues. However, we only quantified murine IFN-gamma expression with our in vivo samples, but expression of human IFN-g was analyzed in vitro. Due to time limitation of the revision, we were not able to obtain the in vivo samples to analyze human IFN-g. We will include these data in our future studies. Thank you.
Q10) Define ECM
A10) As suggested by the Reviewer, we have rectified our mistake and ECM now has been defined on its first use within the Abstract section on page 2 as following; “extracellular matrix (ECM)”
Q11) Line 25: “such as poor availability in a readily usable form due to arduous production”: it is not clear the meaning of the sentence.
A11) We have clarified this sentence in Abstract section on page 2 as following; “In the case of cell-based immunotherapeutics, there are several other bottlenecks preventing translation into clinical use due to their biological nature; for example, poor availability of cell therapeutic in a readily usable form due to difficulties in production, handling, shipping, and storage.”
Q12) Page 2, Line 88-89: “These NK cells induced potent cytotoxic effects against human pancreatic cancer cells in vitro…”. Clarify whether these NK cells are thawed NK cells after cryopreservation.
A12) As suggested by the reviewer, we have clarified information of these NK cells in our revised Introduction section on page 5 as following; “These cryopreserved allogenic NK cells after thawing induced potent cytotoxic effects against human pancreatic cancer cells in vitro …”
Q13) Page 16, Line 333: authors state: “NK cells were also detected in the lungs at all time points, suggesting that systemically administered NK cells can be trapped in the lungs as a “first pass” tissue after systemic injection”. Trapping of cells in the lungs occurs always in mice after i.v. injections of cells, not only for NK cells. Authors should delete that statement.
A13) We have deleted that statement in our revised Result section on page 20. Thank you.
Q14) Page 4, line 144: after TNFa, it appears a strange symbol instead of a brake.
A14) Weird symbol has been rectified in our revised manuscript and it is now correctly displayed as TNF-a. Thank you.
Q15) Figure 2: In the text states that cytotoxicity assays were performed at 1:1 E:T ratio, and in the figure legend it states 10:1 E:T ratio. Which one is it?
A15) We appreciate Reviewer’s valuable comment. We realized how interpretation of our data from Figure 2 in Results section on page 8 might have been misleading, as we did not explain that E:T ratio was different for cytotoxicity assay (E:T ratio = 10:1) and our assessment of degranulation marker/cytokines (E:T ratio = 1:1). We have removed misleading E:T ratio mentioned within Results section and the E:T ratios for both assays are better represented in revised Figure Legend on page 9 as following; “…calcein-AM release assay using K562 and various pancreatic cell lines (AsPC-1, MIA PaCa-2, Capan-1, and PANC-1) at an effector:target (E:T) ratio of 10:1. The levels of degranulation marker CD107a and cytokine (IFN-γ and TNF-α) expression on NK cells were assessed by flow cytometry following co-incubation of K562 or pancreatic cancer cells with NK cells at an E:T ratio of 1:1.”
Q16) Line 381: “Interestingly, these NK cells showed lower expression level of activating receptors after co-cultivation with pancreatic cancer cells”. This phenomenon has already been described as trogocytosis. Authors should include it in the discussion.
A16) We appreciate the valuable Reviewer’s comment. This issue has been addressed in Discussion section on page 24 of our manuscript as following; “…via cell contact-dependent trogocytosis, which may promote immune synapse formation through receptor-ligand interaction [76-78].”
Q17) Figure 3, 5 and 8 are divided in different pages. They need to be edited to fit it into one page each figure.
A17) As recommended by the Reviewer, we have edited the manuscript to have these figures fit into a single page.
The newly revised manuscript also follows the editorial instructions regarding style and format. With these responses and changes, we hope the manuscript is now acceptable for publication in Cancers. Please feel free to contact me if you have any questions.
Sincerely yours,
Chae-Ok Yun, Ph.D.
Professor
Department of Bioengineering
College of Engineering
Hanyang University
17 Haengdang-Dong, Seongdong-Gu, Seoul, Korea
Tel: 82-2-2220-0491
Fax: 82-2-2220-4850
E-mail: chaeok@hanyang.ac.kr

Round 2
Reviewer 1 Report
The authors have addressed all the issues I raised and I think that the manuscript improved substantially.
However, I still believe that including freshly isolated NK cells in their experiments would be important.
The authors justify their choice to exclude this comparison by saying that fresh and thawed NK cells have been thoroughly compared in another article. I do not think that this is satisfying or even appropriate. They should show that this is still the case in their assays and in their lab. Furthermore, since they performed some functional assays that were not included in the previous article (which is then the only novel thing in this manuscript), assuming that fresh NK cells would behave the same because they behaved the same in some other assay in another lab, is rather audacious.
Author Response
June 28, 2019
Professor Danica Jevtic
Assistant Editor, Cancers
Re: Cryopreserved human natural killer cells exhibit potent antitumor efficacy against orthotopic pancreatic cancer through efficient tumor homing and cytolytic ability
Dear Dr. Danica Jevtic,
Thank you for accepting our paper for revision. We fully appreciate the thoughtful reviewer comments as suggested are submitting a revised version of our manuscript, which addresses specific issues brought forth by the reviewer. Detail description of our response and changes are outlined below.
Reviewer 1
Q) The authors have addressed all the issues I raised and I think that the manuscript improved substantially. However, I still believe that including freshly isolated NK cells in their experiments would be important. The authors justify their choice to exclude this comparison by saying that fresh and thawed NK cells have been thoroughly compared in another article. I do not think that this is satisfying or even appropriate. They should show that this is still the case in their assays and in their lab. Furthermore, since they performed some functional assays that were not included in the previous article (which is then the only novel thing in this manuscript), assuming that fresh NK cells would behave the same because they behaved the same in some other assay in another lab, is rather audacious.
A) We agree with the Reviewer that inclusion of freshly isolated NK cells as experimental group in our study would be meaningful. For these type of experiments, freshly isolated NK cells and cryopreserved NK cells should be derived from same donor, as various properties of NK cells can differ from donor to donor basis. GreenCross LabCell cannot obtain freshly isolated NK cells from the same donor as the cryopreserved NK cells used in the present report. Even if donor agreed to donate additional cells to GreenCross LabCell, it would take more than 3 weeks to isolate and expand sufficient NK cell population. Due to time limitation of the revision procedure and difficulty in obtaining cells from same donor, we could not perform these sets of experiments as requested by the Reviewer. Instead, we present the data comparing fresh versus cryopreserved NK cells performed by different collaborators of GreenCross LabCell, who utilized NK cells from same donor as those used in our present report. As shown below, freshly isolated and cryopreserved NK cells exhibited similar cytolytic efficacy, level of granulation marker, and cytokine secretion, even when the experiments were performed by group other than GreenCross LabCell.

Reviewer 2 Report
The authors have replied correctly to the minor changes.
Regarding major changes the authors have looked for published references to justify their results. Even though this could be valid, the main object of this study is to demonstrate the efficacy of cryopreserved NK cells. Therefore, at least one study comparing efficacy of NK cells before and after thawing is required to show consistence of their results.
Regarding the MTT assay, discrimination based on FSC /SSC is not enough, as when tumor cells are dying the size could decrease, at least perform one assay showing appropriate markers.
Other minor changes:
There are many strange symbols, similar to @, along the manuscript. Please, revise them all.
Figure 3: asterisks indicating significant p values are missing.
Results, Section 2.8: “Collectively, these results demonstrate that NK cells efficiently home to tumor tissues and infiltrate them, thus resulting in potent tumor growth inhibition”. In my opinion, finding NK cells after 6 administrations (3 times per week during 2 weeks) and analyzing 1, 2 and 7 days after NK cell administration does not demonstrate a potent tumor growth inhibition. Please, delete “potent”.
Discussion:
Line 372: “likely mediated”, and Line 378: “likely due”. Based on the results presented stating “likely” is too much. Change for “could be due” or “might be due to”
Author Response
June 28, 2019
Professor Danica Jevtic
Assistant Editor, Cancers
Re: Cryopreserved human natural killer cells exhibit potent antitumor efficacy against orthotopic pancreatic cancer through efficient tumor homing and cytolytic ability
Dear Dr. Danica Jevtic,
Thank you for accepting our paper for revision. We fully appreciate the thoughtful reviewer comments as suggested are submitting a revised version of our manuscript, which addresses specific issues brought forth by the reviewer. Detail description of our response and changes are outlined below.
Reviewer 2
The authors have replied correctly to the minor changes. Regarding major changes the authors have looked for published references to justify their results. Even though this could be valid, the main object of this study is to demonstrate the efficacy of cryopreserved NK cells.
Q1) Therefore, at least one study comparing efficacy of NK cells before and after thawing is required to show consistence of their results.
A1) We agree with the Reviewer that inclusion of freshly isolated NK cells as experimental group in our study would be meaningful. For these type of experiments, freshly isolated NK cells and cryopreserved NK cells should be derived from same donor, as various properties of NK cells can differ from donor to donor basis. GreenCross LabCell cannot obtain freshly isolated NK cells from the same donor as the cryopreserved NK cells used in the present report. Even if donor agreed to donate additional cells to GreenCross LabCell, it would take more than 3 weeks to isolate and expand sufficient NK cell population. Due to time limitation of the revision procedure and difficulty in obtaining cells from same donor, we could not perform these sets of experiments as requested by the Reviewer. Instead, we present the data comparing fresh versus cryopreserved NK cells performed by different collaborators of GreenCross LabCell, who utilized NK cells from same donor as those used in our present report. As shown below, freshly isolated and cryopreserved NK cells exhibited similar cytolytic efficacy, level of granulation marker, and cytokine secretion, even when the experiments were performed by group other than GreenCross LabCell.
Q2) Regarding the MTT assay, discrimination based on FSC /SSC is not enough, as when tumor cells are dying the size could decrease, at least perform one assay showing appropriate markers.
A2) As suggested by the Reviewer, we performed FACS analysis to differentiate between cancer and NK cell population at 96 h after co-cultivation using neurotensin receptor 1 (NTR)-specific antibody. Cancer and NK cells were differentiated based on their difference in the expression levels of NTR on the surface of human cancer cells. NTR was recently identified as a useful biomarker for targeted cancer therapy due to its high expression in progressive ductal pancreatic, breast, head and neck cancers (Neuropeptides. 2011;45:151-156). Indeed, flow cytometry revealed that NK cell population were NTR-negative, while 2 pancreatic cancer cell lines (Mia PaCa-2 and PANC-1) were NTR-positive (data not shown). Under the new experimental conditions, NK cells still exerted potent and dose-dependent pancreatic cancer cell killing effect, in a similar manner as to those achieved in Figure 5.
Q3) There are many strange symbols, similar to @, along the manuscript. Please, revise them all.
A3) The weird symbols in the manuscript file present itself after the merging procedure for manuscript into PDF format on the manuscript submission site. We have tried to address this issue in our manuscript, and hopefully the weird symbol has been rectified in our revised manuscript. Thank you.
Q4) Figure 3: asterisks indicating significant p values are missing.
A4) Thank you for pointing our mistake. We have added asterisks indicating significant p values in the revised Figure 3.
Q5) Results, Section 2.8: “Collectively, these results demonstrate that NK cells efficiently home to tumor tissues and infiltrate them, thus resulting in potent tumor growth inhibition”. In my opinion, finding NK cells after 6 administrations (3 times per week during 2 weeks) and analyzing 1, 2 and 7 days after NK cell administration does not demonstrate a potent tumor growth inhibition. Please, delete “potent”.
A5) We have deleted “potent” in our revised Result section on page 20 as following; “Collectively, these results demonstrate that NK cells efficiently home to tumor tissues and infiltrate them, thus resulting in tumor growth inhibition.” Thank you.
Q6) Line 372: “likely mediated”, and Line 378: “likely due”. Based on the results presented stating “likely” is too much. Change for “could be due” or “might be due to”
A6) As recommended by the Reviewer, we have changed “likely due to” to “might be due to” in our revised Discussion section on page 25 as following; “This might be due to gemcitabine reducing the MDSC population….” Thank you.
The newly revised manuscript also follows the editorial instructions regarding style and format. With these responses and changes, we hope the manuscript is now acceptable for publication in Cancers. Please feel free to contact me if you have any questions.
Sincerely yours,
Chae-Ok Yun, Ph.D.
Professor
Department of Bioengineering
College of Engineering
Hanyang University
17 Haengdang-Dong, Seongdong-Gu, Seoul, Korea
Tel: 82-2-2220-0491
Fax: 82-2-2220-4850
E-mail: chaeok@hanyang.ac.kr

Round 3
Reviewer 1 Report
The authors have addressed all my suggestions and comments.
Author Response
July 04, 2019
Professor Danica Jevtic
Assistant Editor, Cancers
Re: Cryopreserved human natural killer cells exhibit potent antitumor efficacy against orthotopic pancreatic cancer through efficient tumor homing and cytolytic ability
Dear Dr. Danica Jevtic,
Thank you for accepting our paper for revision. We fully appreciate the thoughtful reviewer comments as suggested are submitting a revised version of our manuscript, which addresses specific issues brought forth by the reviewer. Detail description of our response and changes are outlined below.
Reviewer 1
The authors have addressed all my suggestions and comments.
The newly revised manuscript also follows the editorial instructions regarding style and format. With these responses and changes, we hope the manuscript is now acceptable for publication in Cancers. Please feel free to contact me if you have any questions.
Sincerely yours,
Chae-Ok Yun, Ph.D.
Professor
Department of Bioengineering
College of Engineering
Hanyang University
17 Haengdang-Dong, Seongdong-Gu, Seoul, Korea
Tel: 82-2-2220-0491
Fax: 82-2-2220-4850
E-mail: chaeok@hanyang.ac.kr

Reviewer 2 Report
Minor: Figure 3 has not been modified in the manuscript, just in the reviewer's comments. Please, modify it in the manuscript.Author Response
July 04, 2019
Professor Danica Jevtic
Assistant Editor, Cancers
Re: Cryopreserved human natural killer cells exhibit potent antitumor efficacy against orthotopic pancreatic cancer through efficient tumor homing and cytolytic ability
Dear Dr. Danica Jevtic,
Thank you for accepting our paper for revision. We fully appreciate the thoughtful reviewer comments as suggested are submitting a revised version of our manuscript, which addresses specific issues brought forth by the reviewer. Detail description of our response and changes are outlined below.
Reviewer 2
Q1) Minor: Figure 3 has not been modified in the manuscript, just in the reviewer's comments. Please, modify it in the manuscript.
A1) Thank you for pointing our mistake. We have added the revised Figure 3 in our revised manuscript.
The newly revised manuscript also follows the editorial instructions regarding style and format. With these responses and changes, we hope the manuscript is now acceptable for publication in Cancers. Please feel free to contact me if you have any questions.
Sincerely yours,
Chae-Ok Yun, Ph.D.
Professor
Department of Bioengineering
College of Engineering
Hanyang University
17 Haengdang-Dong, Seongdong-Gu, Seoul, Korea
Tel: 82-2-2220-0491
Fax: 82-2-2220-4850
E-mail: chaeok@hanyang.ac.kr
